# Significant Enhancement of Hydrogen-Sensing Properties of ZnO Nanofibers through NiO Loading

**DOI:** 10.3390/nano8110902

**Published:** 2018-11-03

**Authors:** Jae-Hyoung Lee, Jin-Young Kim, Ali Mirzaei, Hyoun Woo Kim, Sang Sub Kim

**Affiliations:** 1Department of Materials Science and Engineering, Inha University, Incheon 402-751, Korea; jhlee5321@naver.com (J.-H.L.); piadote@naver.com (J.-Y.K.); 2The Research Institute of Industrial Science, Hanyang University, Seoul 133-791, Korea; alisonmirzaee@yahoo.com; 3Department of Materials Science and Engineering, Shiraz University of Technology, Shiraz 7155713876, Iran; 4Division of Materials Science and Engineering, Hanyang University, Seoul 133-791, Korea

**Keywords:** ZnO, NiO loading, *p*-*n* heterojunction, nanofiber, gas sensor, sensing mechanism

## Abstract

Metal oxide *p*-*n* heterojunction nanofibers (NFs) are among the most promising approaches to enhancing the efficiency of gas sensors. In this paper, we report the preparation of a series of *p*-NiO-loaded *n*-ZnO NFs, namely (1−x)ZnO-xNiO (x = 0.03, 0.05, 0.7, 0.1, and 0.15 wt%), for hydrogen gas sensing experiments. Samples were prepared through the electrospinning technique followed by a calcination process. The sensing experiments showed that the sample with 0.05 wt% NiO loading resulted in the highest sensing performance at an optimal sensing temperature of 200 °C. The sensing mechanism is discussed in detail and contributions of the *p*-*n* heterojunctions, metallization of ZnO and catalytic effect of NiO on the sensing enhancements of an optimized gas sensor are analyzed. This study demonstrates the possibility of fabricating high-performance H_2_ sensors through the optimization of *p*-type metal oxide loading on the surfaces of *n*-type metal oxides.

## 1. Introduction

Among various available gas sensors, metal oxides are one of the most promising gas sensors for sensing hydrogen gas [1]. This stems from their advantages in terms of high sensitivity and stability, low fabrication costs, and simplicity of use [2]. In particular, gas sensors fabricated from one-dimensional (1D) nanostructured metal oxides have become very popular in recent years based on their small size and large surface area, which results in numerous adsorption sites for target gases and facilitates rapid diffusion of gas molecules [3]. Furthermore, 1D nanostructures can be easily synthesized at low cost through simple procedures [4]. Therefore, many researchers have studied the gas sensing characteristics of 1D nanostructured metal oxides [5,6].

ZnO is one of the most promising *n*-type oxides for gas sensing, based on its large band gap, high chemical and thermal stability, low price, high mobility of charge carriers, and the intrinsic sensitivity of ZnO [7,8,9]. Accordingly, ZnO-based gas sensors with different morphologies have been reported for sensing of different gases. For example Barreca et al. used urchin-like ZnO nanosrods for detection of different reducing and oxidizing gases [10]. Comini et al. used single crystal ZnO nanowires for room temperature NO_2_ gas sensing [11]. Barreca et al. used 1D assembly of ZnO nanostructures for detection of H_2_ and CH_4_ gases [12]. Singh et al. used In_2_O_3_-ZnO core-shell nanowires (NWs) for detection of NO_2_ gas [13]. Waclawik and co-workers prepared organic functionalized ZnO nanowires for detection of toxic gases [14]. Comini et al. used ZnO NWs for detection of ethanol gas [15].

Among the different ZnO nanostructures utilized for sensing purposes, ZnO nanostructures with nanofiber (NF) morphology are very popular. In addition to their high surface area, they contain ultrafine nanograins, which further increase the number of adsorption sites for target gases [16,17,18]. Another advantage of NFs is their simple and inexpensive synthesis through the electrospinning technique [19].

In addition, ZnO in form of NFs benefit from high surface area due to NF morphology. For hydrogen sensing in particular, the metallization effect of ZnO (where the outer surfaces of ZnO are reduced to metallic zinc with high conductivity) in an H_2_ atmosphere can significantly affect the sensing properties of gas sensors [20]. However, for increasingly strict accuracy standards, ZnO-based gas sensors with better performance are required. One popular method to increase the sensitivity of ZnO NF gas sensors is to create n-p heterojunction NFs by adding a *p*-type metal oxide to an *n*-type ZnO [21]. For *p*-*n* heterojunctions, based on the transfer of electrons from *n*-ZnO to a *p*-type metal oxide, the resistance of the gas sensor is high in air. Upon exposing such a sensor to hydrogen gas, a large change in sensor resistance will lead to a stronger response compared to a pure ZnO NF sensor. Furthermore, through the injection of H_2_ gas, metallization of ZnO surfaces can occur, resulting in a semi-conductor-to-metallic conversion. Therefore, the heterojunctions between *n*-ZnO and *p*-type metal oxides are destroyed, which changes the resistance of a gas sensor [20,22]. Accordingly, sensors fabricated from ZnO NFs loaded with a *p*-type metal oxide are expected to show excellent hydrogen sensing performance.

Among available *p*-type metal oxides, NiO is particularly important for sensing purposes based on its excellent structural stability, high tendency toward the adsorption of oxygen, and strong catalytic activity [23]. Therefore, it has been extensively investigated for sensing of different gases [24,25]. Through the combination of *n*-ZnO and *p*-NiO, it is possible to develop gas sensors with superior sensing performance. For example, Liu et al. demonstrated the superior acetone sensing performance of ZnO-NiO nanocomposites compared to pure ZnO [26]. Deng et al. reported the superior ethanol sensing properties of ZnO-NiO composites compared to pure NiO sensors [24].

However, most researchers have simply studied the gas sensing properties of one particular composition without performing any compositional optimization. Because the final gas response strongly depends on the composition of a gas sensor, it is necessary to explore the gas-sensing properties of a series of compositions. To address this issue, in this study, we prepared three NiO-loaded ZnO NF gas sensors with different amounts of NiO (0.03, 0.05, 0.07, 0.1, and 0.15 wt%) to study the effects of composition on the hydrogen sensing behavior of the resultant gas sensors. Hydrogen is a promising energy source based on several advantages, including renewability, abundance, efficiency, and environmental friendliness. However, it is highly explosive in concentrations above 4 vol% in air. Therefore, the fabrication of high-quality gas sensors for H_2_ sensing is urgent [27,28]. Hydrogen gas sensing experiments revealed that the sensor with 0.05 wt% NiO loading had the strongest gas response. The strong response of the optimized NiO-loaded ZnO NF gas sensor was mainly caused by the metallization effect of ZnO and formation of *n*-ZnO/*p*-NiO heterojunctions. This study demonstrates the need for compositional optimization to achieve the strongest possible gas response in *p*-type-loaded *n*-type metal oxides.

## 2. Materials and Methods

### 2.1. Preparation of the Solution for Electrospinning

The synthesis of viscous solution of for preparation of ZnO-NiO NFs was based on our previous papers [29,30]. Analytical grades of polyvinyl alcohol (PVA, MW = 80,000), zinc chloride dihydrate (ZnCl_2_·2H_2_O), and nickel (II) acetate tetrahydrate (Ni(CH_3_COO)_2_·4H_2_O) were provided by Sigma-Aldrich (St. Louis, MO, USA). Deionized (DI) water was utilized as a solvent for the preparation of all solutions. First, PVA was added to DI water and stirred (400 rpm) at 70 °C for 4 h to create a 10 wt% PVA aqueous solution. Subsequently, 1 g of ZnCl_2_·2H_2_O and calculated amounts (0.01 wt% to 0.15 wt%) of Ni precursor were added (0.05 mL/h) to the polymer solution and stirred (400 rpm) for 12 h at 70°C. The final viscous solutions (280 mPa·s) for electrospinning were then prepared. The amount of Ni^2+^ precursor was changed to prepare solutions with different nominal compositions of NiO (0.03, 0.05, 0.07, 0.1, and 0.15 wt%) for the final samples.

### 2.2. Electrospinning

The electrospinning solution was loaded into a plastic syringe with a metallic needle with inner diameter of 0.051 mm. To initiate and accelerate electrospinning, a large positive voltage (+15 kV) and large negative voltage (−10 kV) were applied to the needle and Al collector, respectively. During electrospinning, the distance between the tip of the needle and collector was fixed as 0.2 m and the feed rate was set to 0.01 mL/h. The as-synthesized NFs were annealed at 600 °C for 2 h to remove the polymer and water to obtain crystalline phases.

### 2.3. Material Characterization

The morphological features of the synthesized products were studied through field-emission scanning electron microscopy (FE-SEM, Hitachi-S-4200, Hitach, Ltd., Tokyo, Japan). Additional details of microstructural features were studied through transmission electron microscopy (TEM) with a JEOL JEM-3010 (JEOL, Ltd., Tokyo, Japan). Elemental analysis was performed utilizing an energy-dispersive X-ray spectrometer (EDS, JEOL, Ltd., Tokyo, Japan) incorporated in the TEM.

### 2.4. Gas Sensing Measurements

The sensing experiments in this study were similar to those described in our previous paper [31]. To fabricate gas sensors, Ti (~50 nm thickness) and Pt (∼200 nm thickness) electrodes were sequentially magnetron sputter-deposited onto the synthesized NFs that had been deposited on the SiO_2_-grown Si substrates (200 nm thickness). The spacing between electrodes was 300 μm. The fabricated gas sensors were put into a gas chamber (2.5 × 2.2) inside a horizontal type quartz furnace, with possibility of temperature control automatically. Sensing tests were performed utilizing a gas dilution and testing system. The gas concentrations were precisely controlled by varying the mixture ratio of the target gas in N_2_ as carrier gas and synthetic air (20% O_2_, 80% N_2_) by utilizing mass flow controllers (MFCs). Total flow rate of gases was set to 500 sccm before and after injection of target gas. In other words, the flow rate was equal when the chamber was exposed to air and target gases to avoid any possible variation in the sensing properties. Additionally, due to limited volume of gas chamber and close distance of MFCs with gas chamber, the introduction of target gas to gas chamber takes very short time, which is much faster than the response times of gas sensors, therefore, we can ignore these effects on the dynamic behavior of gas sensor.

The sensing properties of the gas sensors were investigated in the presence of H_2_ gas as the main target gas and the selectivity of the optimized sensor was tested in the presence of CO and C_6_H_6_ gases at the optimized temperature of 200 °C. Since oxygen gas has a significant effect on the gas sensing response, it was constant during the measurements, including introduction of synthetic air and subsequent injection of target gas. As an example, based on the total flow rate of 100 sccm, for introduction of a pre-determined concentration of hydrogen gas, the relative amounts of N_2_ and H_2_ gas (100 ppm in N_2_ background) were changed according to Table 1.

The resistances of the gas sensors in air (R_0_) and in the presence of the target gas (R_g_) were recorded utilizing a Keithley source meter (Keithley Instrument Ltd. 2400 model, Cleveland, OH, USA). Sensor’s response was calculated as S (%) = (R_0_ − R_g_)/R_0_ × 100 = ΔR/R_0_ × 100. Response and recovery times were defined as the time required for the change in the resistance to reach 90% of the final value after the injection and stoppage of H_2_ gas, respectively. The sensing behaviors were studied under various relative humidity (RH = 0–79.4%, measured at 25 °C) levels. Figure 1 shows an illustration of the synthesis steps and preparation of the gas sensors.

## 3. Results and Discussion

### 3.1. Morphological and Structural Analyses

Figure 2a presents FE-SEM images of the as-spun NFs prior to calcination. They had a smooth morphology similar to nanowires (NWs) and without any nanograins. However, after calcination as a result of the evaporation of the solvent and polymer, the surfaces became rough. Figure 2b–d present FE-SEM images of 0.97ZnO-0.03NiO, 0.95ZnO-0.05NiO, and 0.90ZnO-0.1NiO NFs, respectively. They clearly reveal formation of NFs with lengths up to several micrometers. The insets in these figures show ultrafine grains on the surfaces of the synthesized NFs. This is a unique feature of NFs that discriminates NFs from NWs. These nanograins play the role of double Schottky barriers to electron transfer, which enhance the gas sensing performance of the sensors. The diameters of the NFs is in the range of 125–150 nm. The fabricated NFs have a consistent diameter because the calcination temperature was the same for all samples and the amount of loaded NiO was not significantly different between the samples.

Figure 3a presents a typical TEM micrograph of ZnO-NiO NFs with diameters of approximately 200 nm. Figure 3b presents a lattice-resolved TEM micrograph. Interplanar distances of 0.25 nm and 0.24 nm can be indexed to the (101) and (111) planes of ZnO and NiO, respectively, demonstrating the coexistence of crystalline ZnO and NiO crystals in the synthesized NFs. Figure 3c–e present EDS color-mapping analyses of 0.95ZnO-0.05NiO NFs, confirming the coexistence of Zn, Ni, and O elements.

### 3.2. Gas Sensing Properties

The sensor with the 0.97ZnO-0.03NiO composition was exposed to 1, 5, and 10 ppm H_2_ gas at various temperatures (50–300 °C) in 50 °C steps to find the optimal working temperature. Figure 4 presents the dynamic resistance of the 0.97ZnO-0.03NiO sensor at different sensing temperatures.

As it can be seen, upon exposure to H_2_ gas at all temperatures, the resistance decreases. Upon the stoppage of H_2_ and introduction of air, the resistance returns to its original value. Since H_2_ gas is a reducing gas, this behavior demonstrates the *n*-type nature of sensor, resulting from the *n*-type semiconducting nature of ZnO, which is the major constituent of the proposed NiO-loaded ZnO NF gas sensor. Furthermore, the reversibility of the sensor is excellent, meaning its signal returns to the air value after the reintroduction of air. Figure 5a plots the responses of the 0.97ZnO-0.03NiO sensor to 10 ppm H_2_ gas versus sensing temperature. This plot shows a peak at 200 °C, where the maximum response was observed for 10 ppm H_2_ gas. At lower temperatures, the activation energy for the adsorption of H_2_ is insufficient. At higher temperatures, the rate of gas desorption is much higher than the adsorption rate, meaning the gas escapes prior to effective adsorption on the surface of sensor [32].

Figure 5b shows the decrease in the initial resistance of the sensor with increasing sensing temperature. This behavior is caused by the fact that with an increase in temperature, more electrons in the valence band of ZnO exit to the conduction band, meaning the conductivity increases. The response and recovery times of the gas sensor at different temperatures in the presence of 10 ppm H_2_ are presented in Figure 6. It should be noted that due to use of highly accurate MFCs for introduction and stoppage of target gas and synthetic air, it was possible to calculate response and recovery times precisely. The gas sensor shows relatively fast dynamic times because of its 1D nanostructure. This morphology facilitates rapid mass transfer of H_2_ molecules to and from the inner regions of the sensor and improves the speed at which charge carriers can traverse the barriers along the NFs [33]. When increasing the sensing temperature, both the response and recovery times decrease because of faster gas diffusion.

To find the optimal composition for NF sensors, 0.03, 0.05, 0.7, 0.1, and 0.15 wt% NiO-loaded ZnO NF sensors were exposed to 1, 5, and 10 ppm H_2_ gas at 200 °C, which was identified as the optimal temperature. Figure 7 presents dynamic resistance plots of the sensors when they were exposed to H_2_ gas. The sensors did not show a meaningful response to 1 ppm H_2_ gas, but responses to 5 ppm and 10 ppm H_2_ gas are clearly visible. Figure 8 illustrates the dependency of the responses on the composition of the sensors. With an increase in NiO loading from x = 0.03 to 0.05 wt%, the sensing performance increases. With a further increase in the amount of NiO to 0.1 wt%, the response decreases.

The selectivity of the optimized sensor (0.05 wt% NiO-loaded ZnO) was analyzed by exposing the sensor to CO and C_6_H_6_ gases. Figure 9a presents the dynamic resistance curves of the gas sensor towards CO and C_6_H_6_ gases, as well as the curve for H_2_ gas. As shown in the selectivity pattern presented in Figure 9b, the sensor demonstrates the highest response to H_2_ gas, followed by the C_6_H_6_ and CO gases.

To see if the gas sensor can detect lower concentration of H_2_ gas, it was exposed to 0.1–10 ppm H_2_ gas and dynamic resistance curves are presented in Figure 10a. As shown in Figure 10b it can detect even very low concentrations of H_2_ gas, i.e, 0.1 ppm H_2_ gas. It should be noted that the thermal conductivity coefficient of hydrogen (174 mW/(m⋅K) at 298 K, 1 atm pressure) is much higher that than of air under the same conditions (26 mW/(m⋅K) [34]. Therefore, when the hydrogen gas introduced into the gas-sensing chamber the temperature of the sensor can be changed and the resistance of gas sensor can be changed which causes an uncertainty in the sensing results. In this regard, Bierer et al. reported effect of hydrogen gas on the rising of temperature during the gas sensing measurements [34]. They reported that for 400 ppm hydrogen gas, a temperature change of 3.6 K was observed, and for a 40 ppm hydrogen gas it was decreased to 0.8 K. Even though each gas system and sensing material has its unique feature, at low concentrations the temperature rise can be neglected. Since in our measurements maximum hydrogen concentration was set to 10 ppm, so we can neglect the temperature rise due to presence of hydrogen gas with high thermal conductivity coefficient.

Since in real applications humidity is always present in the environment, we tested the response of 0.05 wt% NiO-loaded ZnO NF gas sensor to 10 ppm H_2_ gas under different values of relative humidity (RH%) as shown in Figure 11a. As shown in Figure 11b, with increasing of RH%, the response decreases. This is due to the fact that water molecules will be adsorbed on the surfaces of gas sensor and will prevent the chemisorption of oxygen species and target gas species. Therefore, they will decrease the available sites for adsorption of H_2_ gas, leading to a lower response of gas sensor in the presence of RH [35].

### 3.3. Gas Sensing Mechanism

When a metal-oxide-based gas sensor is exposed to a target gas, its resistance changes. By measuring this change, the nature and concentration of the target gas can be determined. First, when the NiO-loaded ZnO NF sensor is in an air environment, the oxygen gas in air is adsorbed on the surface of the sensor [36,37,38,39]:

O_2(g)_ → O_2(ads)_(1)


(2)$$O_{2(\mathrm{ads})}+e^{-}\to O_{2}^{-}{}_{\left(\mathrm{ads}\right)}$$

(3)$$O_{2}^{-}+e^{-}\to 2O^{-}{}_{\left(\mathrm{ads}\right)}$$


O^−^ + e^−^ → O^2−^(4)

O^2−^_(ads)_ is typically introduced at high temperatures, whereas the other oxygen species are stable at lower temperatures [40]. In general, $O_{2}^{-}$, O^−^, and O^2−^ are stable at a temperature <150 °C, 150–400 °C, and >400 °C, respectively [39]. Therefore, at a sensing temperature of 200 °C, the dominant oxygen species can be assumed to be O^−^.

According to above reactions, the oxygen species can extract electrons from the conduction band of ZnO, which results in the creation of an electron-depletion layer on the outer surfaces of ZnO in air. Additionally, in NiO-loaded ZnO NFs, there are two different types of semiconducting materials in which the majority of carriers are electrons (in ZnO) and holes (in NiO). When they are in direct contact, the electrons in ZnO flow into NiO to equate their Fermi levels (Figure 12a). Accordingly, the width of the depletion layer in ZnO NFs increases compared to pure ZnO and results in a higher resistance for the NiO-loaded sensor compared to pure ZnO. Therefore, it can be concluded that in ZnO-NiO composite NFs, there are two types of depletion layers: one is created by the abstraction of electrons by oxygen gas in the air and another is created by the formation of *p*-NiO/*n*-ZnO heterojunctions. When the sensor is in an H_2_ atmosphere, hydrogen molecules react with adsorbed oxygen ions on the surface of the gas sensor (Figure 12b) according to the following equation [41]:

H_2_ + O^−^ → H_2_O + *e*^−^(5)


Additionally, because of the catalytic characteristics of NiO, it can dissociate H_2_ molecules into H atoms (H_adsorbed_), which spill over onto the ZnO surface to react with adsorbed oxygen species (O^−^) according to following equations [42,43]:

H_2_ → 2H
(6)



2H_(ads)_ + O^−^ → H_2_O + *e^−^*(7)

As a result of above reactions, the released electrons return to the surface of sensor, which results in a large decrease in the width of the electron depletion layer on the sensor. Consequently, a response can be observed. Additionally, based on the abundance of ZnO nanograins, there are significant *n*-ZnO-*n*-ZnO homojunctions that create potential barriers to the flow of electrons. Modulation of the potential height at homojunctions in the presence of H_2_ alters the resistance, which eventually contributes to the appearance of a sensor signal. However, it should be noted that because there are very low amounts of NiO in the fabricated sensors, it is unlikely that *p*-NiO/*p*-NiO homojunctions significantly affect the final sensing performance of the sensors.

Furthermore, because of the strong reducing effect of hydrogen, the surfaces of the ZnO at the sensing temperature may be partially reduced to metallic Zn. Metallization of the ZnO surface occurs based on the adsorption of H_2_ atoms on the O sites of the nonpolar surfaces of ZnO. Charge delocalization occurs between Zn and the O–H bonds, and partially occupies the 4s and 3d states of the surface Zn atoms [20]. This surface metallization causes a resistance change in the surfaces of ultrafine ZnO nanograins and contributes to the sensor response (Figure 12a). When H_2_ gas is removed and air is supplied, the metallic Zn recovers to ZnO, thereby reestablishing the original band configuration as well as original baseline resistance.

The similar percolation phase transitions were also previously reported. For example, Kneer et.al. used CuO for detection of H_2_S, where upon exposing to H_2_S at high temperature, CuS exhibited metallic conductivity, in which the outer layers underwent a percolation phase transition and the resistivity significantly decreased [44,45]. Since the percolation time was a function of H_2_S concentration, this method could be used to monitor the H_2_S content of the gas mixture in a quasi-continuous fashion because the individual percolation events were statistically distributed as a function of the H_2_S concentration. Similarly, in our present study, the metallization effect making the sensing mechanism selective by inducing significant resistance changes upon metallization transition.

Next, we determined why the sensor with the optimized composition (i.e., 0.05 wt% NiO-loaded sensor) showed the highest response to H_2_ gas. It should be noted that all synthesized NFs were calcined at the same temperature, meaning it is assumed that all sensors have nearly the same morphology, surface area and grain sizes. Therefore, we excluded these factors in our analysis. This means that the number of *p*-NiO/*n*-ZnO heterojunctions and amount of NiO must significantly affect the sensing results. For the 0.97ZnO-0.03NiO composition, the small number of ZnO-NiO heterojunctions will result in the weakest sensing performance. In the 0.95ZnO-0.05NiO composition, the response was enhanced by the increased number of ZnO-NiO heterojunctions. Furthermore, this composition had higher amounts of NiO and oxygen ions, which could be easily adsorbed on the surface of the NiO because Ni^2+^ ions are easily oxidized into Ni^3+^ (Figure 12c) [32]. This results in a greater resistance change for the gas sensor. However, for the compositions with NiO > 0.5 wt%, the response was decreased. This is likely a due to (i) weaker gas sensing properties of *p*-type NiO in comparison with *n*-type ZnO (ii) possible decrease of ZnO-NiO heterojunctions due to increase of NiO-NiO homojunctions, in fact, instead of being in direct contact with the *n*-ZnO grains, the *p*-NiO nanograins established direct contact with each other, resulting in the degradation of sensor response.

## 4. Conclusions

A series of NiO-loaded ZnO NFs were fabricated through the electrospinning technique followed by calcination. Characterization results verified the formation of NiO-loaded ZnO NFs with the desired morphology and composition. Gas sensing results at the optimal sensing temperature of 200 °C demonstrated that the sensor response decreased when the amount of NiO was greater than 0.05 wt%, indicated that the amount of NiO must be optimized to achieve superior sensing properties. The optimized gas sensor also showed good selectivity to H_2_ gas in the presence of C_6_H_6_ and CO gases. Also effect of humidity was studied, where in the presence of water molecules, the response was decreased. The excellent sensing performance of the optimized sensor can be attributed to the formation of *p*-*n* heterojunctions, metallization of ZnO by H_2_, and catalytic effect of NiO towards H_2_ gas. This study demonstrates the necessity for optimization of *p*-type loading on *n*-type NFs to obtain superior sensing properties.

## Figures and Tables

**Figure 1 nanomaterials-08-00902-f001:**
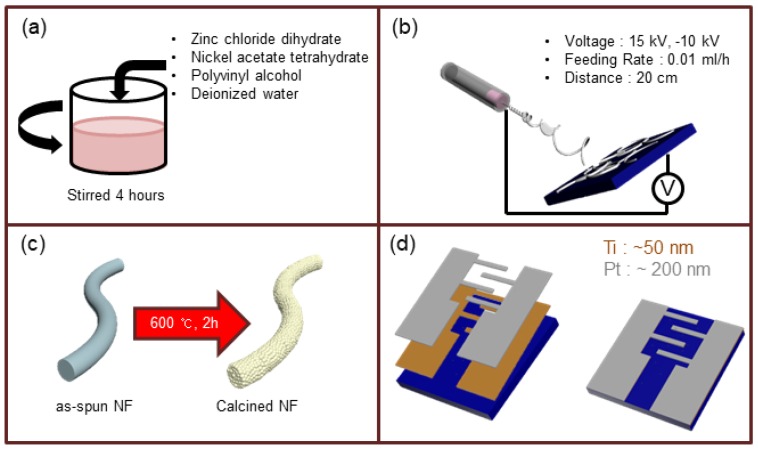
Schematic illustration of preparation steps for the synthesis of NiO-loaded ZnO NFs: (**a**) Preparation of viscous solution for electrospinning, (**b**) electrospinning procedure, (**c**) calcination of synthesized NFs, and (**d**) sensor fabrication.

**Figure 2 nanomaterials-08-00902-f002:**
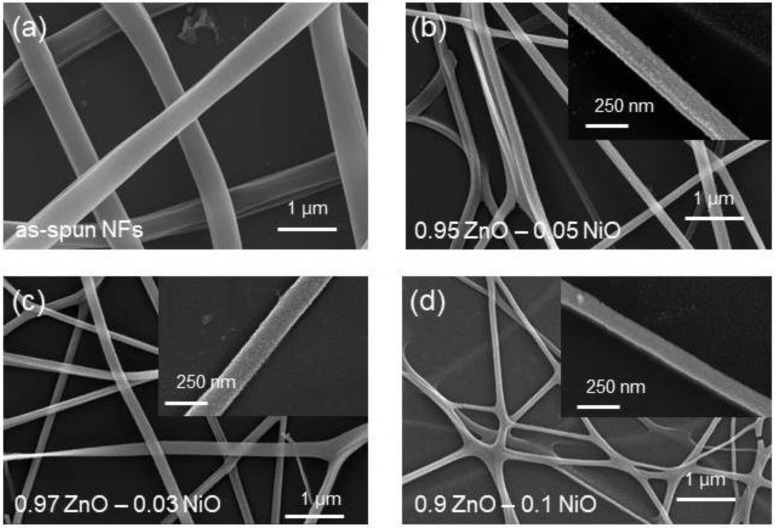
FE-SEM images: (**a**) 0.03 wt% NiO-loaded ZnO NFs before calcination, (**b**) 0.03 wt% NiO-loaded ZnO NFs after calcination, (**c**) 0.05 wt% NiO-loaded ZnO NFs after calcination, and (**d**) 0.1 wt% NiO-loaded ZnO NFs after calcination. Insets show corresponding magnified FE-SEM images.

**Figure 3 nanomaterials-08-00902-f003:**
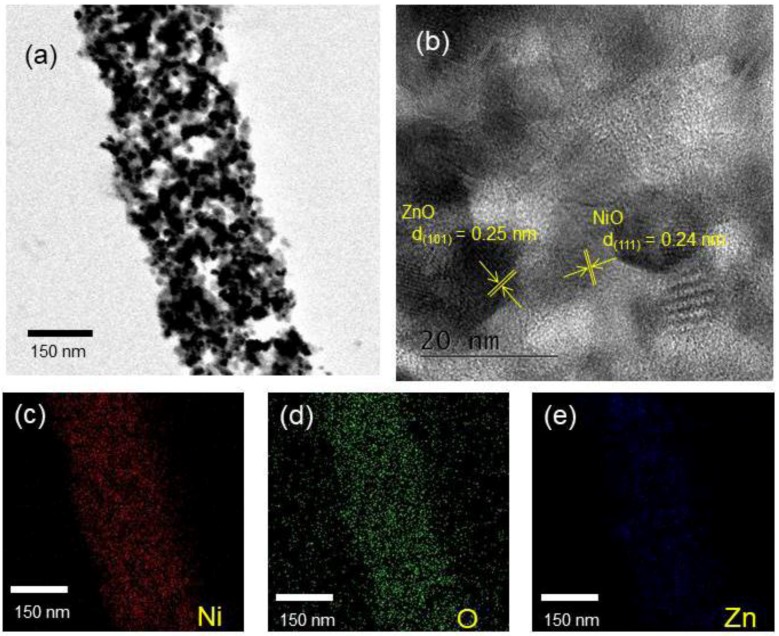
(**a**) Typical high-magnification TEM image of ZnO-NiO composite NFs, (**b**) corresponding lattice-resolved TEM image showing the lattice fringes of ZnO and NiO, and (**c**)–(**e**) EDS color-mapping analyses of 0.05 wt% NiO-loaded ZnO NFs.

**Figure 4 nanomaterials-08-00902-f004:**
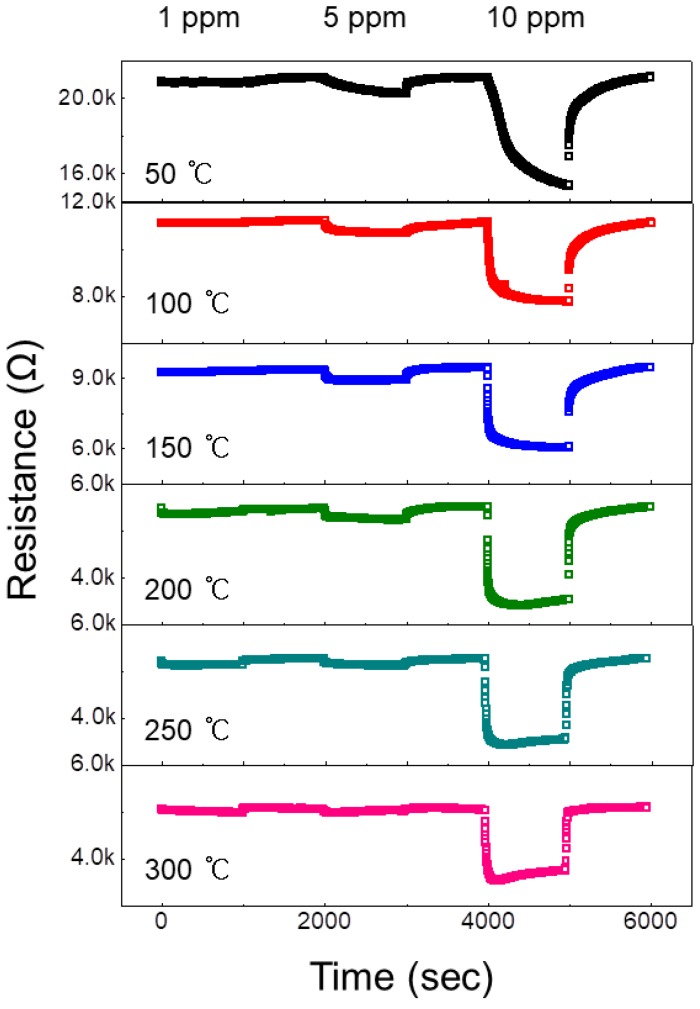
Dynamic resistance curves of 0.05 wt% NiO-loaded ZnO NF sensor for 1, 5 and 10 ppm H_2_ gas at different sensing temperatures.

**Figure 5 nanomaterials-08-00902-f005:**
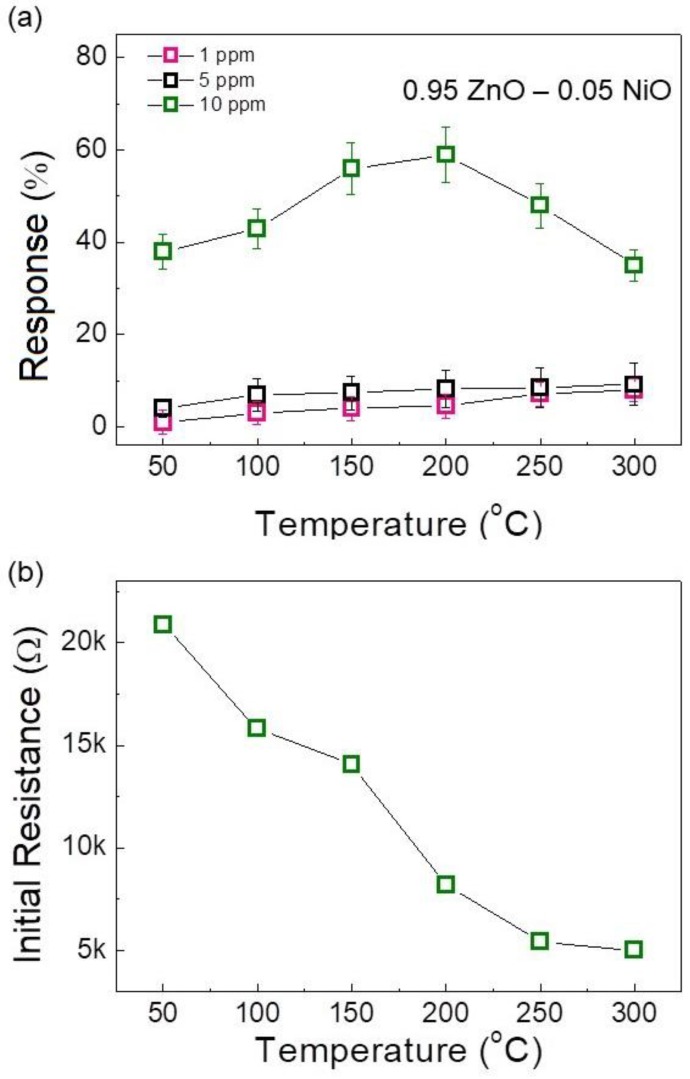
(**a**) Dependence of H_2_ gas response on sensing temperature for 0.05 wt% NiO-loaded ZnO NF sensor; and (**b**) initial resistance of 0.05 wt% NiO-loaded ZnO NF sensor as a function of sensing temperature.

**Figure 6 nanomaterials-08-00902-f006:**
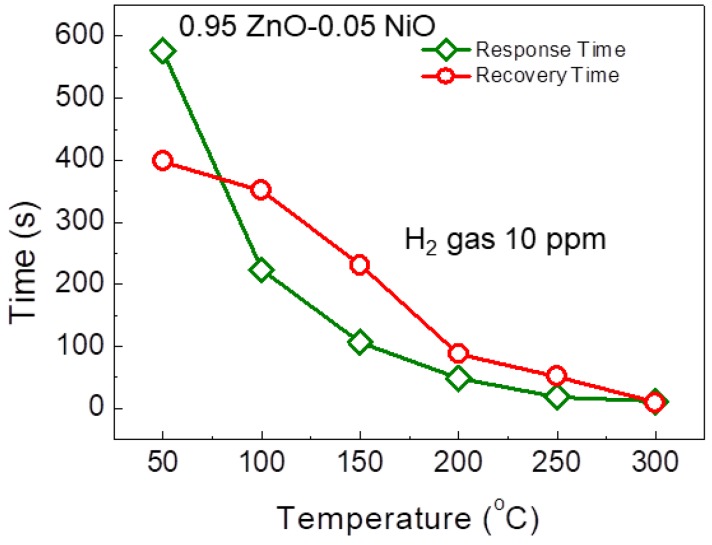
Response and recovery times of 0.05 wt% NiO-loaded ZnO NF sensor.

**Figure 7 nanomaterials-08-00902-f007:**
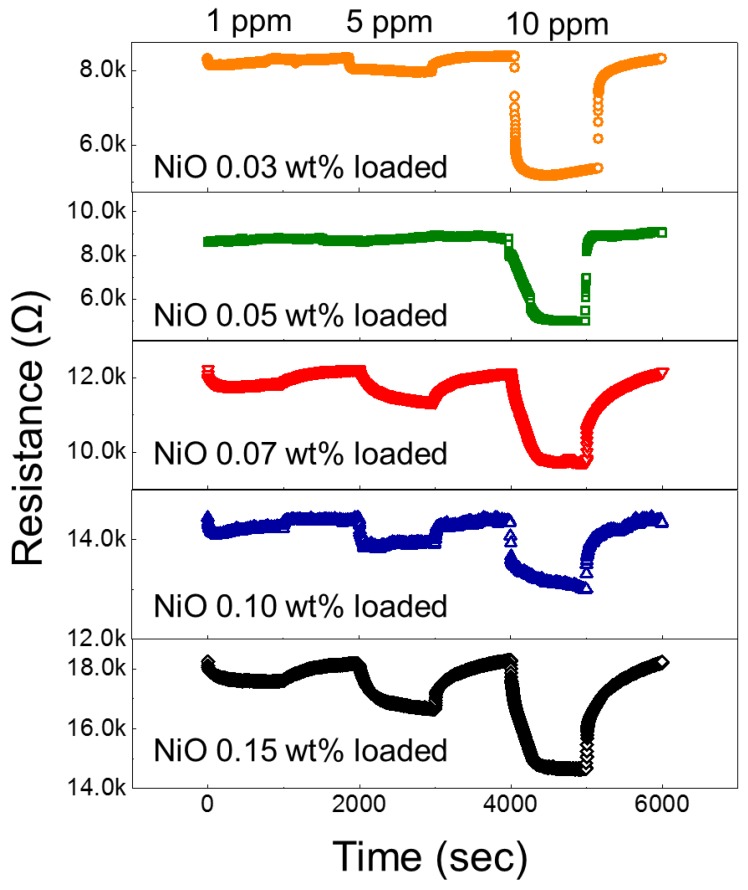
Dynamic normalized resistance curves of different sensors for various concentrations of H_2_ gas at an optimal sensing temperature of 200 °C.

**Figure 8 nanomaterials-08-00902-f008:**
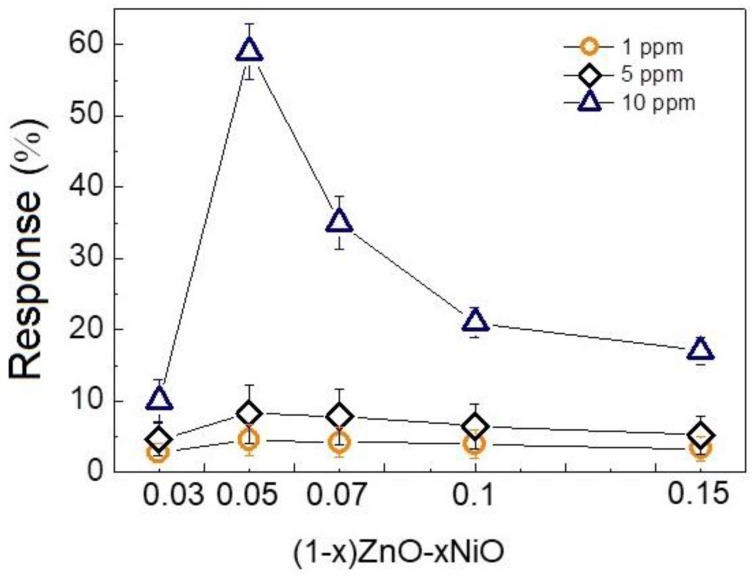
The responses of ZnO-NiO gas sensor for different sensor compositions.

**Figure 9 nanomaterials-08-00902-f009:**
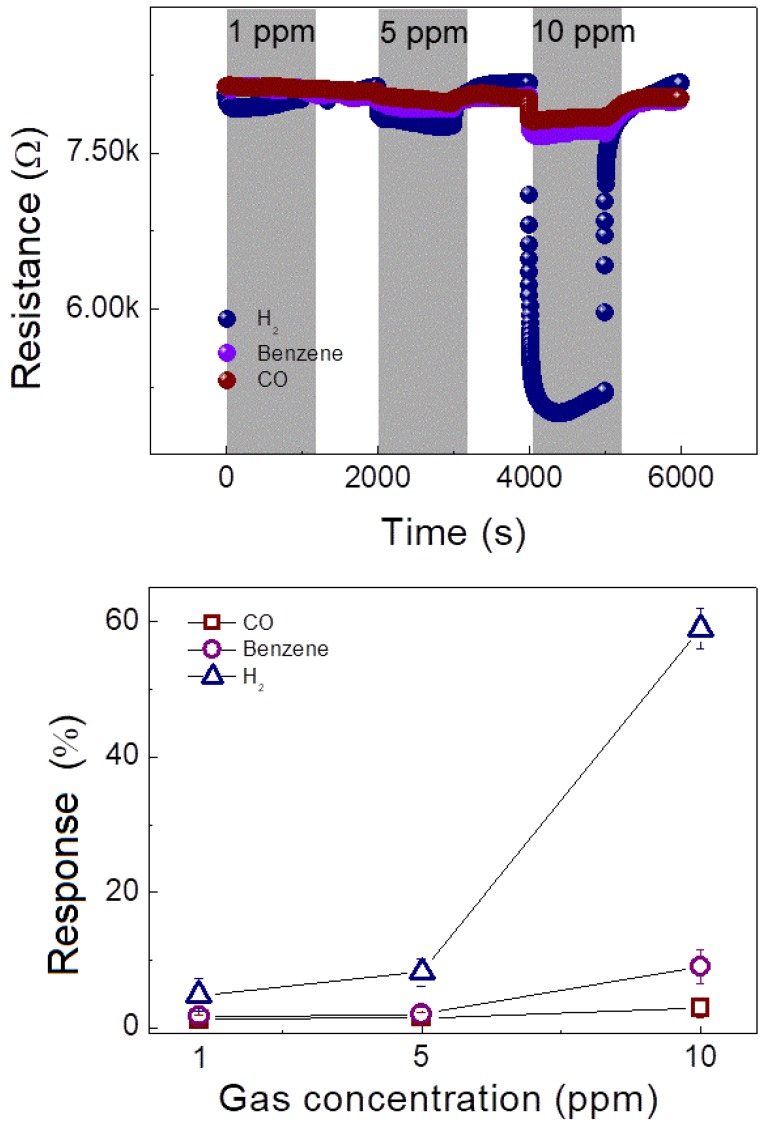
(**a**) Dynamic resistance curves of 0.05 wt% NiO-loaded ZnO NF gas sensor for 1, 5, and 10 ppm concentrations of H_2_, CO, and C_6_H_6_ gases at 200 °C; and (**b**) corresponding response versus gas concentration.

**Figure 10 nanomaterials-08-00902-f010:**
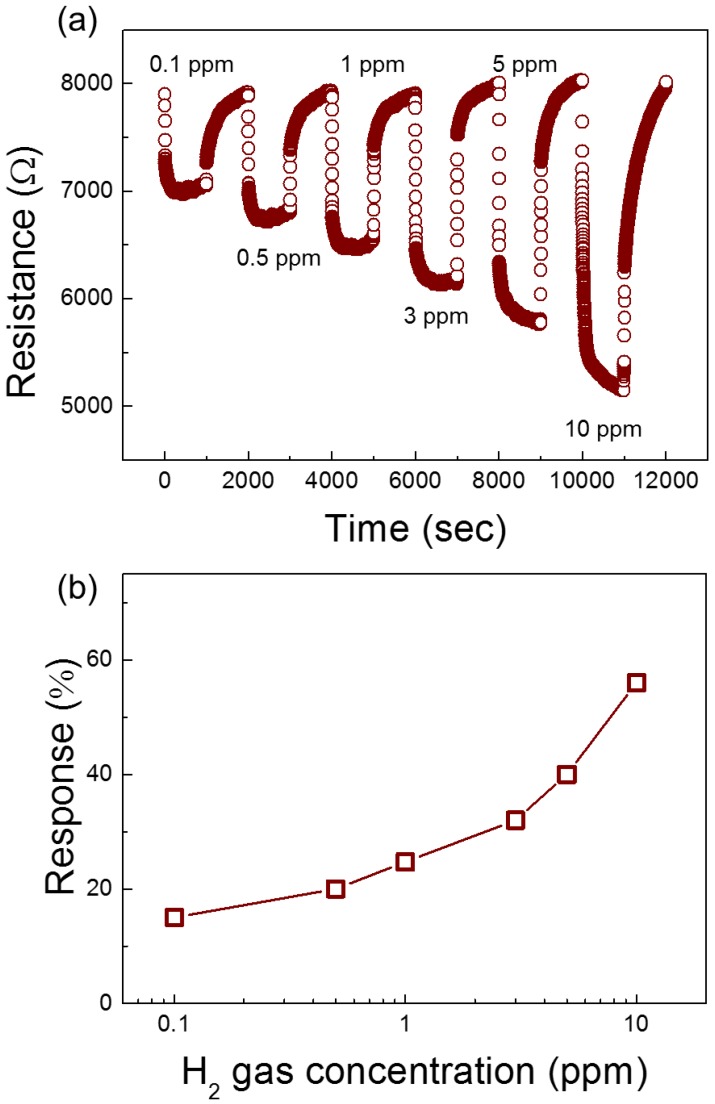
(**a**) Dynamic resistance curve of 0.05 wt% NiO-loaded ZnO NF gas sensor towards 0.1–10 ppm H_2_ gas; and (**b**) the response versus H_2_ gas concentration.

**Figure 11 nanomaterials-08-00902-f011:**
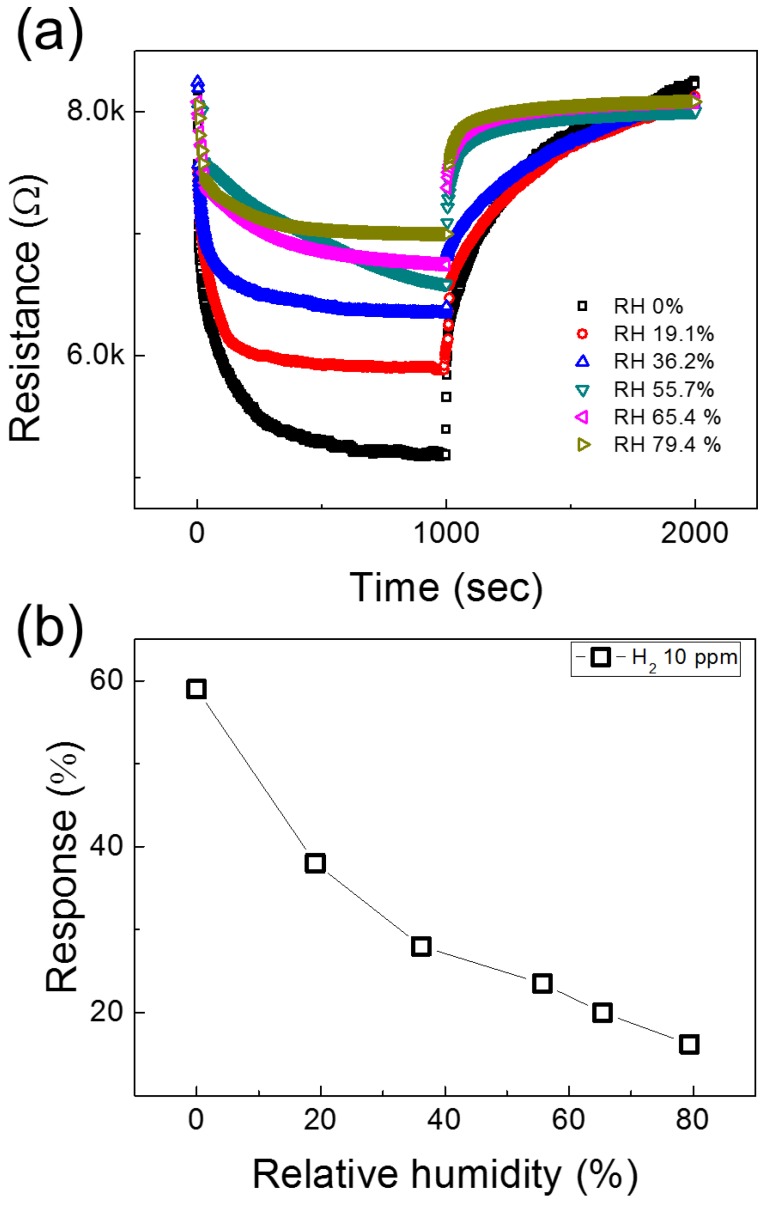
(**a**) Dynamic resistance curves of 0.05 wt% NiO-loaded ZnO NF gas sensor to 10 ppm H_2_ gas in the presence of 0–79.4% RH; (**b**) Response to 10 ppm H_2_ versus RH%.

**Figure 12 nanomaterials-08-00902-f012:**
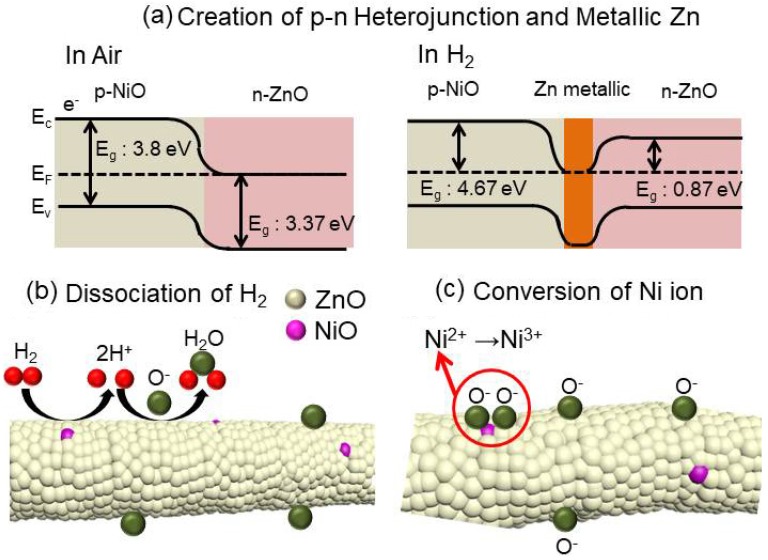
Schematic diagram of sensing mechanism: (**a**) creation of *p*-*n* heterojunctions and metallization of ZnO, and (**b**) dissociation and subsequent oxidation of H_2_ gas on the surface of gas sensor, and (**c**) conversion of Ni ions.

**Table 1 nanomaterials-08-00902-t001:** The procedure for obtaining of different concentrations of hydrogen gas during the gas measurements.

Reference (Synthetic Air)		Target Gas			Final Concentration
O_2_	N_2_	O_2_	N_2_	H_2_ (Concentration 100 ppm in N_2_)	
20 mL/min	80 mL/min	20 mL/min	79 mL/min	1 mL/min	1 ppm
20 mL/min	80 mL/min	20 mL/min	75 mL/min	5mL/min	5 ppm
20 mL/min	80 mL/min	20 mL/min	70 mL/min	10 mL/min	10 ppm

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
