# Peer review of "Significant Enhancement of Hydrogen-Sensing Properties of ZnO Nanofibers through NiO Loading"

_nanomaterials, 2018, doi:10.3390/nano8110902_

Round 1
Reviewer 1 Report
The authors have greatly improved the manuscript and addressed almost all points raised in the first round of reviews. However, some issues need further clarification:
Authors state that the gas mixture is changed by varying the mixture of gas changing the flow. As per your description you use 1 channel with dry synthetic air and 1 channel with H2 in N2.
Now here comes the problem with this: If you reduce the synthetic air flow and add H2 in N2 you reduce the O2 content of your gas sample. This has an enormous effect on the conductivity of your layer and consequently you measure two things at once: less oxygen (decreasing the resistivity) and more H2 (decreasing the resistivity). Please check your experimental setup and make sure that your oxygen content remains constant when changing the H2 concentration. Amend the manuscript accordingly.
Your flow rate is 500 cm³/min. Your chamber has a volume (you do not state a volume) XX cm³. This means that upon changing the gas flow the gas volume of the chamber has to be exchanged 3 times to reach a new steady state gas concentration, i.e. 3 x volume/500 cm³/min. Please discuss the implications on the reaction speed of your sensors.
Authors should add the following information:
Which Keithley did you use?
At what temperature did you measure the r.H.? State the temperature such that one can calculate the absolute water content in air.
Fig. 4: Add numbers to the x-axis and show the set points of H2 concentration to make an interpretation of the reaction easier.
Always add gas composition, i.e. if you have used dry synthetic air.
3.2 Gas sensing properties: The first three sentences belong to the introduction.
The way you discuss percolation as a surface effect in gas sensing is highly linked to the following papers:
Kneer et.al. New method to selectively determine hydrogen sulfide concentrations using CuO layers, SNB, 222 (2016) 625 – 631
Kneer et.al. Specific, trace gas induced phase transition in copper(II)oxide for highly selective gas sensing, Appl. Phys. Lett. 105 (2014) 073509
Please discuss your findings in relation to the findings in these papers, which describe a highly similar surface system.
Last point: Are the authors sure they can neglect the influence of the high thermal conductivity of H2 on the sensor response? See Bierer et.al. MEMS based metal oxide sensor for simultaneous measurement of gas induced changes of the heating power and the sensing resistance, Microsys. Technol. 22 (7) (2016) 1855-1863
Author Response
We upload the replies to the reviewers' comments.

Reviewer 2 Report
Dear Authors,
the paper is clear and the work is complete, thus, in my opinion, it can be accepted for publication in Nanomaterials, even if the degree of novelty of rather limited.
I would suggest minor corrections:
* Line 45: "...the number adsorption sites...", please change in: "...the number of adsorption sites...",
* Lines 88 & 97: "mL" is the correct abbreviation for milliliters, not "ml",
* In Fig 5a, Fig. 8, Fig. 9b: the plots mention "Sensitivity" on ordinates axis. Check it, please: the sensitivity is the slope of the curve sensor response = f(gas concentration), as per IUPAC definition. You plotted the sensor response, please change the caption on ordinates axis in these 3 figures,
* Lines 220-222: "Since in real applications always there are some humidity in the environment, we tested the response of 0.05 wt% NiO-loaded ZnO NF gas sensor to 10 ppm H2 gas in the presence of different amounts of relative humidity (RH%) as shown in Figure 11a.", please change in:
"Since in real applications humidity is always present in the environment, we tested the response of 0.05 wt% NiO-loaded ZnO NF gas sensor to 10 ppm H2 gas under different values of relative humidity (RH%) as shown in Figure 11a.", or something equivalent...,
* Line 236: Check the species generated at low temperature, it is not written correctly.
* Line 236: "...are stable at<150 400="" and="">400 °C,...", please change in: "...are stable at a temperature<150 400="" and="">400 °C,...",
* line 257: "alters" instead of "alter",
* Line 265, 266: To help Readers, you could add another comment on the reversibility of the metallisation effect, as you did in ref [20]: "When H2 gas is removed and air is supplied, the metal Zn recovers to ZnO, thereby reestablishing the original band configuration.",
* Line 273: "result" instead of "resulted".
Best regards.
Reviewer
Author Response

(The authors gave the same response as above.)
